

# Enhancing bullfrog farming sustainability: circular water management through effluent treatment

Dandan Xie[1], Jiehua Hu[1], Liru Lin[1], Xiaomei Huang[1], Changsheng Xie[2] and Haibin He[1]

[1] School of Marine Biology, Applied Technology Engineering Center of Fujian Provincial Higher Education for Marine Resource Protection and Ecological Governance, Xiamen Key Laboratory of Intelligent Fishery, Xiamen Ocean Vocational College, Xiamen, China

[2] Zhangzhou Haizhiwei Biotechnology Co., Ltd., Zhangzhou, China

## ABSTRACT

Ensuring effective wastewater treatment is crucial for promoting sustainable water use and reducing environmental pollution in intensive bullfrog aquaculture. This study presents the design and implementation of an integrated treatment system, composed of a flotation tank, a biochemical tank, and constructed wetlands, aimed at facilitating the reuse of treated effluent. The system, operated under optimal conditions—at a flow rate of 250 m³/h, hydraulic retention time of 6 h, and aeration intensity of 2,000 m³/h—demonstrated significant removal efficiencies. Specifically, the biochemical tank reduced chemical oxygen demand (CODCr), ammoniacal nitrogen, and total phosphorus by 70%, 43%, and 42%, respectively. After a month of continuous operation, the system achieved higher removal rates of 71.7% for CODCr, 83% for ammoniacal nitrogen, and 86.7% for reactive phosphorus, rendering the treated water suitable for reuse in bullfrog farming. However, total nitrogen removal remained relatively low, and reactive phosphorus slightly exceeded discharge standards, indicating areas for further optimization. Despite these limitations, this innovative system enhances water recycling, supports circular water management strategies, and provides a practical solution for reducing water consumption and minimizing aquaculture's environmental footprint.

## INTRODUCTION

The American bullfrog (*Rana catesbeiana*), originally from several eastern U.S. states, has since been introduced to the western U.S. and other countries. In China, it was first imported from Cuba and Japan in 1959 and has become a significant part of the country's aquaculture sector in recent years (*Zhao, 2022*). By 2022, domestic production had risen to 700,000 tons, with breeding operations spanning over 10 provinces and supporting millions of jobs (*Di et al., 2022*). The bullfrog industry is now valued at nearly 80 billion yuan. Known for its high protein (19.9 g) and low fat (0.3 g) per 100 g, bullfrogs have gained popularity among consumers. Its farming is highly efficient, with a feed conversion

Corresponding author
Dandan Xie,
xiedandan@xmoc.edu.cn

ratio of 1.0 (compared to 2.0 for golden pomfret, 1.5 for grass carp, and over 2.5 for pigs and chickens), making it a resource-efficient species. With yields of 150 tons per hectare, bullfrog farming offers high economic returns. The industry shows strong market demand, high profitability, and low resource requirements in terms of water, land, and materials (*Lan et al., 2022*).

There has long been a conflict between the expansion of aquaculture and environmental protection, with two key issues: (1) intensive farming consumes large amounts of water, resulting in significant tailwater discharge that depletes groundwater, and (2) tailwater contains pollutants, including feed residue, waste from excretion, dead organisms, and even leftover medications used during cultivation. These pollutants, including organic matter, nitrogen ($NH_4^+$-N, $NO_3^-$-N, and $NO_2^-$-N), phosphorus ($PO_4^{3-}$-P), and suspended solids, can exist in dissolved, colloidal, or particulate forms. Uncontrolled discharge of these substances into natural water bodies causes environmental damage (*Cai et al., 2013*; *Shi & Lin, 2015*). The discharge of untreated shrimp farm tailwater into the sea led to a significant reduction in nitrite concentrations within 0.5 km and phosphate levels after 1 km, while nitrate and ammoniacal nitrogen remained elevated even at 1.5 km. Untreated tailwater also increased suspended solids and organic matter in receiving waters. Due to its severe pollution issues, bullfrog farming has been prohibited in some regions, resulting in both consumer dissatisfaction and economic losses for farmers. Beyond pollution concerns, bullfrog farming is also restricted in some regions due to its ecological risks, including the discharge of nutrient-rich wastewater contributing to eutrophication and the increased risk of disease transmission to wild populations.

In contemporary China, managing aquaculture production in a way that is both environmentally sustainable and responsible remains a significant challenge. Finding a solution that satisfies all stakeholders—administrators, academics, farmers, and consumers—has become a key issue (*Boyd et al., 2020*; *Liu et al., 2021*; *Shi & Lin, 2015*). Effective treatment and disposal of aquaculture wastewater are critical prerequisites for the industry's sustainable growth. Researchers and scientists have started investigating discharge standards for aquaculture tailwater, and several treatment strategies have shown positive results in various aquaculture operations (*Boyd et al., 2020*; *Liu et al., 2021*; *Tan, Liu & Zhang, 2021*). The concept of "recirculating aquaculture systems" has also been introduced in modern intensive shrimp ponds (*Chang et al., 2020*; *Rajesh, Kumar & Pandey, 2024*). These systems include drains or sumps to collect solids, anaerobic settling zones for denitrification to reduce nitrate and organic matter buildup, and aerobic ponds where species like tilapia graze on algae. After treatment, the water is reused in the shrimp ponds. Additionally, aquatic plants and algae are commonly employed to lower nitrogen and phosphorus levels in aquaculture wastewater (*Ansari et al., 2017*; *Enduta et al., 2011*; *Gichana et al., 2019*; *Nakphet, Ritchie & Kiriratnikom, 2017*; *Prabhath et al., 2022*; *Shan et al., 2016*; *Wang et al., 2022*). While the treatment of wastewater from farmed fish, shrimp, and shellfish has been widely studied, research on the treatment of bullfrog aquaculture effluent remains limited. Moreover, the treatment of bullfrog aquaculture wastewater is largely limited to standalone constructed wetlands (*de Freitas Borges & Tavares, 2017*; *Sipaúba-Tavares, Silva Peres & Scardoeli-Truzzi, 2019*) and

anaerobic-aerobic filtration systems (*Mello et al., 2016*), with little discussion on treatment processes and water reuse in large-scale bullfrog farming operations. Although constructed wetlands have been shown to effectively remove nutrients from wastewater in smaller-scale systems (*Gichana et al., 2019*), their use as a standalone solution in large-scale aquaculture operations remains uncertain due to challenges in scalability and efficiency. Similarly, anaerobic-aerobic filtration systems have been explored in bullfrog farming, offering some benefits in water recirculation and wastewater treatment (*Mello et al., 2016*). However, these systems often face limitations in terms of operational optimization and the ability to handle large volumes of wastewater effectively.

In contrast, this study examines a bullfrog farming facility in Zhangzhou, Fujian Province, China, covering an area of two hectares, as a case study. Taking into account the facility's unique terrain features, a comprehensive tailwater treatment system was designed, incorporating flotation tanks, biochemical tanks, and constructed wetlands. Unlike standalone constructed wetlands or other biological treatment methods, which face challenges in large-scale applications, this integrated system provides a more efficient and scalable solution. Through continuous refinement of the system's operating conditions, wastewater treatment efficiency was significantly improved, enabling effective water reuse in bullfrog farming. Unlike traditional methods, our system enhances nitrogen and phosphorus removal through a sequential treatment process, improving water quality while enabling effluent reuse in bullfrog farming. This study not only demonstrates a viable approach to enhancing sustainability in bullfrog aquaculture, but also offers a more robust framework for integrating water reuse and reducing environmental impact in large-scale operations. The findings provide valuable insights for the development of similar systems in other aquaculture facilities, contributing to more sustainable, resource-efficient farming practices that address the limitations.

## MATERIALS AND METHODS

### Experimentation site

A bullfrog breeding facility located in Zhangzhou, Fujian Province, was chosen as the experimental site. Spanning a total area of 3.4 hectares, the farm includes a 2-hectare water body dedicated to bullfrog farming. The entire process, from breeding and hatching to rearing tadpoles and young frogs into adulthood, was carried out on-site. Both groundwater and recycled treated tailwater were utilized in the farming operations. Portions of this text were previously published as part of a preprint (*Xie et al., 2023*).

### Tailwater treatment system construction

The tailwater treatment system was composed of a flotation tank, a primary biochemical tank, a secondary biochemical tank, and constructed wetlands (Fig. 1). The treated effluent is not discharged; instead, it is reintroduced into the bullfrog breeding facility for reuse. Sampling was conducted at the outlets of each treatment stage (sampling points 1–5). Key water quality indicators, including chemical oxygen demand (CODCr), ammoniacal nitrogen, and total phosphorus, were selected to assess the effectiveness of each treatment process. These parameters were measured in accordance with the national standards

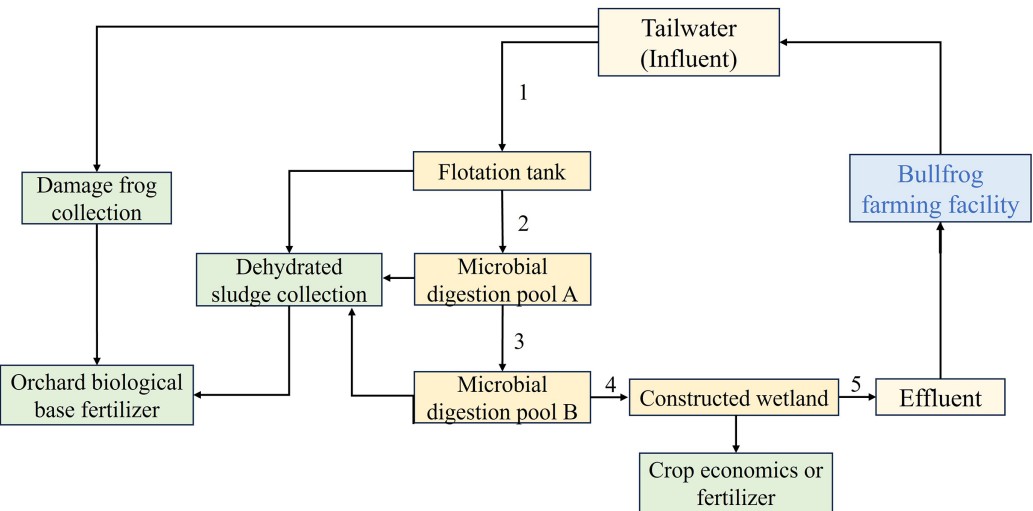

**Figure 1 Treatment procedure for bullfrog aquaculture tailwater.** 1–5 are sampling points.

outlined in the *Water and Wastewater Monitoring and Analysis Methods* (4th Edition with Supplementation) (*State Environmental Protection Administration, 2002*). Removal efficiency = (Ci–Co)/Ci × 100%, where Ci = concentration of inlet and Co = concentration of outlet.

The tailwater was first directed into the flotation tank for aeration. During this stage, suspended solids rose to the surface, forming a scum layer that was skimmed off using a strainer. After this, the tailwater was transferred to the inlet of the flotation tank (sampling point 1). Once flotation pretreatment was completed, the water proceeded to the biochemical tank system. This system consisted of two interconnected canvas tanks, each with a capacity of 1,500 $m^3$, acting as the primary and secondary biochemical tanks. Micropore aeration, driven by turbines, was used at the tank bottoms to maintain a dissolved oxygen level of at least 2 mg/L. The effluent from the secondary tank was then directed to the constructed wetland before being discharged. The three components of this treatment system are depicted in Fig. 2.

A series of 10 interconnected cement tanks (20 m long × 10 m wide × 0.6 m high) were arranged in a cascading manner to allow water flow along the gradient. The effluent from the biochemical tank was directed into these tanks. Watermifoil (*Myriophyllum verticillatum L.*) was planted in three tanks, water spinach (*Ipomoea aquatica Forsk*) in four tanks, and alligator weed (*Alternanthera philoxeroides* (Mart.) Griseb.) in three tanks (Fig. 3). These three aquatic plant species were selected based on preliminary experiments evaluating their adaptability to the tailwater conditions. The results showed that they exhibited strong growth performance, effectively absorbing nutrients and contributing to water quality improvement. In contrast, other tested species, such as purslane (*Portulaca oleracea*) and water cress (*Enhydra fluctuans*), demonstrated poor growth under the same conditions and were therefore not included in the system. Each tank held around 100 $m^3$ of water, with a total combined capacity of 1,000 $m^3$ across the 10 tanks. Water flowed

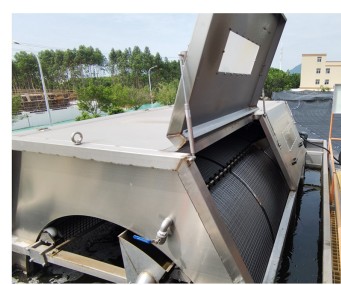

Flotation tank

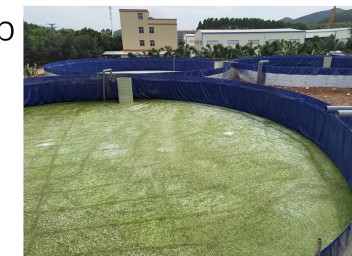

Microbial digestion pool

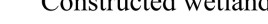
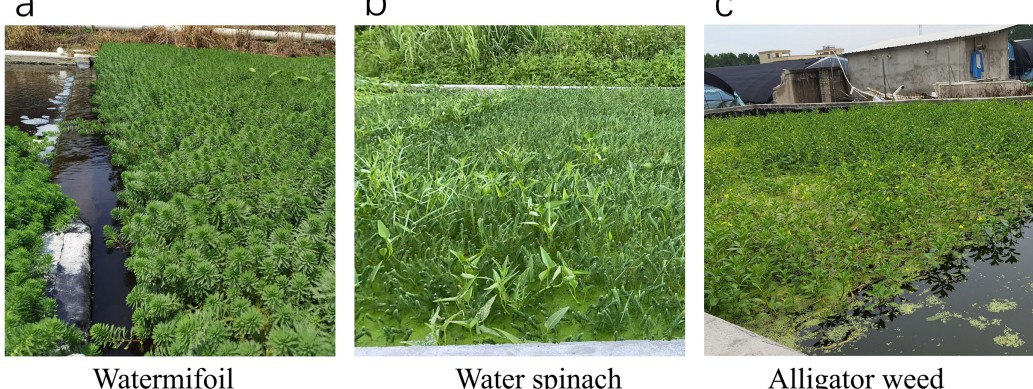

Constructed wetland

**Figure 2 The three units of the treatment system: flotation tank (A), microbial digestion pool (B), and constructed wetland (C).**

Watermifoil

Water spinach

Alligator weed

**Figure 3 Different plants in constructed wetland systems: watermifoil (A), water spinach (B), and alligator weed (C).**

through the cascade for 4–5 h, and the outlet of the cascade marked the endpoint of the tailwater treatment system (sampling point 5). The treated effluent was then channeled into a purified water tank, allowing it to be reused in breeding operations as necessary.

## Biochemical tank operating condition optimization

To achieve better treatment results, we optimized the conditions of the biochemical treatment unit, including the hydraulic retention time and aeration intensity. The aeration intensity was maintained at 2,000 $m^3$ $h^{-1}$ while the inflow rate ($m^3$ $h^{-1}$) and corresponding

**Table 1 Influent flow rate and hydraulic retention time of the biochemical tank.**

| Factor | Gradient setting | | | | |
|---|---|---|---|---|---|
| Influent flow rate (m$^3$/h) | 400 | 300 | 250 | **200***  | 100 |
| Hydraulic retention time (h) | 3.75 | 5 | **6*** | 7.5 | 15 |

**Note:**
* The bold numbers with asterisks represent the optimal process conditions.

hydraulic retention time (h) were adjusted (Table 1). Samples were collected at sampling point 4 for $COD_{Cr}$, ammoniacal nitrogen, and total phosphorus determination.

Based on the results of hydraulic retention time (see Results, Fig. 3A), an inflow rate of 250 m$^3$ h$^{-1}$ and hydraulic retention time of 6 h were set for optimizing aeration intensity. Under these conditions, the aeration intensity was varied (500, 1,000, 1,500, 2,000, and 3,000 m$^3$ h$^{-1}$). Samples were collected at sampling point 4 for $COD_{Cr}$, ammoniacal nitrogen, and total phosphorus determination.

## Evaluation of tailwater treatment system performance

The effluent generated from intensive bullfrog aquaculture was treated using the established wastewater treatment system. The purified water, stored in the treated water tank, was reused in breeding as necessary. After a month of continuous operation, water samples were taken from two locations: the system's inlet (sampling point 1) and the outlet (sampling point 5). The parameters of water quality, along with the corresponding analytical methods, were based on national standards outlined in the *Water and Wastewater Monitoring and Analysis Methods* (4th Edition, with Supplement) by *State Environmental Protection Administration (2002)* (see Table 2). The removal efficiency was calculated using the formula: Removal = (Ci − Ce)/Ci × 100%, where Ci represents the concentration of the pollutant in the influent and Ce represents the concentration in the effluent after treatment.

Under the optimized operating conditions for the biochemical tank (refer to Results, Fig. 3), the system was set to function with an inflow rate of 250 m$^3$/h, a hydraulic retention time of 6 h, and an aeration rate of 2,000 m$^3$/h. After initiating the operation, samples were taken at five designated points (1–5) to measure the levels of CODCr, ammoniacal nitrogen, and total phosphorus, allowing an assessment of the treatment efficiency of each unit in the system. To evaluate the system's overall performance, the water quality of the effluent was compared to the Grade 1 standard specified in the Integrated Wastewater Discharge Standard (GB 8978-1996), as set by the State Environmental Protection Administration.

## Data processing

Data were expressed as mean ± standard deviation (SD). The statistical analysis was conducted using SPSS version 26.0. A one-way analysis of variance was used to examine the difference between treatment outcomes under different operating conditions. Multi-way analysis was performed either with least significant difference (LSD) (homogeneous variance) or Games-Howell (heterogeneous variance). Two-sample $t$-tests

**Table 2 Water quality parameter and measurement methods.**

| No. | Parameter | Measurement method* |
|---|---|---|
| 1 | pH | pH meter |
| 2 | Suspended solids | Gravimetry |
| 3 | Dissolved oxygen | Iodometry |
| 4 | Chemical oxygen demand ($COD_{Cr}$) | Dichromate method |
| 5 | Biochemical oxygen demand ($BOD_5$) | 5-day biochemical oxygen demand test |
| 6 | Ammoniacal nitrogen (measured as the amount of N) | Nessler's reagent spectrophotometry |
| 7 | Nitrite nitrogen (measured as the amount of N) | Sulfanilamide/N-(1-naphthyl)-ethylenediamine dihydrochloride spectrophotometry |
| 8 | Total nitrogen (measured as the amount of N) | Persulphate oxidation |
| 9 | Reactive phosphorus (measured as the amount of P) | Phosphomolybdenum blue spectrophotometry |
| 10 | Total phosphorus (measured by the amount of P) | Persulphate oxidation |
| 11 | Color intensity | Serial dilution |
| 12 | Odor intensity | Qualitative analysis (cold method) |

Note:
*According to the *Water and Wastewater Monitoring and Analysis Methods* (4th Edition with Supplementation), P. R. China: Environmental Protection Administration (*State Environmental Protection Administration, 2002*).

were used to analyze the physiochemical difference between the influent and effluent of the treatment system. Graphs and tables were plotted/drawn with Excel 2010.

# RESULTS AND DISCUSSION

## Effect of hydraulic retention time and aeration intensity on biochemical tank performance

Due to the significant impact of hydraulic retention time (HRT) on microbial activity and pollutant removal efficiency, optimizing HRT enhances nutrient degradation while maintaining system stability. The impact of hydraulic retention time on treatment performance is illustrated in Fig. 4A. CODCr removal improved as retention time increased, reaching 70% at a 6-h retention time, 74% at 7.5 h, and 80% at 15 h. Similarly, ammoniacal nitrogen removal also rose with longer retention times, achieving 43% removal at 6 h, though further increases in retention time showed little additional effect. Total phosphorus removal followed a similar trend, peaking at 42% at 6 h before slightly decreasing with extended retention times. Based on treatment efficiency, processing time, and energy consumption for all three pollutants, the optimal hydraulic retention time for the biochemical tank was determined to be 6 h, with a corresponding influent flow rate of 250 m³/h.

Figure 4B shows the effect of aeration intensity on treatment outcomes. CODCr removal improved with increasing aeration intensity, peaking at 2,000 m³/h, after which removal efficiency began to decline. The impact of aeration intensity on ammoniacal nitrogen removal was minimal, with a slight decrease in removal efficiency at higher aeration rates (2,500 m³/h). For total phosphorus, removal increased with aeration intensity up to 1,500 m³/h, after which further increases provided only marginal improvement. Taking into account treatment effectiveness, time, and energy usage, the ideal aeration intensity for the biochemical tank was set at 2,000 m³/h.

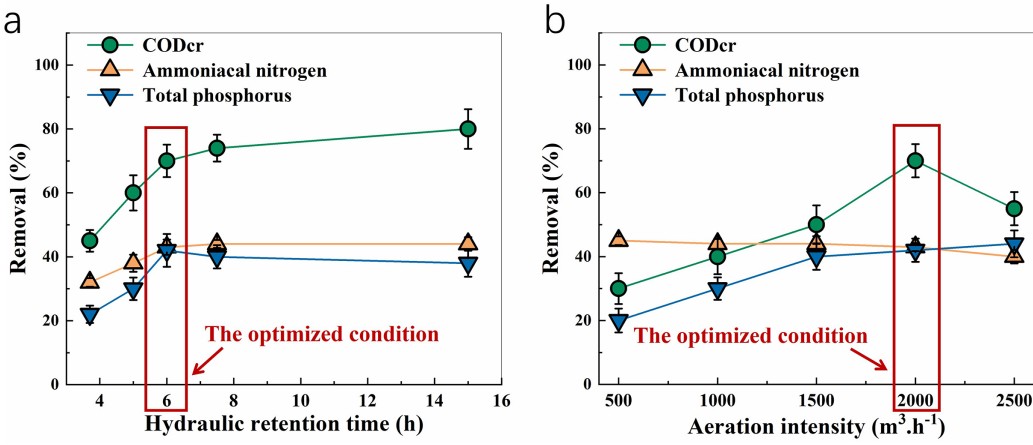

**Figure 4 Effect of hydraulic retention time (A) and aeration intensity (B) on the removal of $COD_{Cr}$, ammoniacal nitrogen, and total phosphorus.**

## Effect and efficiency of each treatment system unit

As shown in Fig. 5A, $COD_{Cr}$ concentration in tailwater was reduced from 120 mg·L$^{-1}$ (sampling point 1) to 110 mg·L$^{-1}$ (sampling point 2) after the flotation tank treatment, which resulted in a removal of only 8.3%. This means that the flotation tank had little removal effect on $COD_{Cr}$. After the aeration process of the primary biochemical tank, $COD_{Cr}$ concentration decreased to 76 mg·L$^{-1}$ (sampling point 3), and further dropped to 36 mg·L$^{-1}$ (sampling point 4) after the aeration process of the secondary biochemical tank. The removal of the two tanks totaled 70%, indicating effective $COD_{Cr}$ removal by these tanks. The $COD_{Cr}$ level of biochemical tank effluent was reduced to 34 mg·L$^{-1}$ as the water moved through the wetlands (sampling point 5), a removal of only 5.6% from the previous step. The efficiency of $COD_{Cr}$ removal was in the order of biochemical tank >> flotation tank > wetlands.

Figure 5B shows that after passing through the flotation tank, the ammoniacal nitrogen level in tailwater dropped from 26.1 mg·L$^{-1}$ (sampling point 1) to 25.0 mg·L$^{-1}$ (sampling point 2), a mere 4.2% removal. This means that the flotation tank had little removal effect on ammoniacal nitrogen. After the treatment of both biochemical tanks, the ammoniacal nitrogen level was reduced to 14.9 mg·L$^{-1}$ (sampling point 4), a 42.9% removal. The ammoniacal nitrogen level in the effluent was further reduced to 4.4 mg·L$^{-1}$ (sampling point 5) after going through the wetlands, a removal of 83.1%. The efficiency of ammoniacal nitrogen removal was in the order of wetlands >> biochemical tank >> flotation tank.

Figure 5C shows that after going through the flotation tank, the total phosphorus level in tailwater decreased from 34.0 mg·L$^{-1}$ (sampling point 1) to 28.3 mg·L$^{-1}$ (sampling point 2), a 16.8% removal. The total phosphorus level after treatment in the biochemical tanks was further reduced to 19.7 mg·L$^{-1}$ (sampling point 4), a removal of 42.1%. As the biochemical tank effluent moved through the wetlands, the total phosphorus was reduced to 8.0 mg·L$^{-1}$ (sampling point 5) in the water present in the wetlands, a removal of 76.5%.

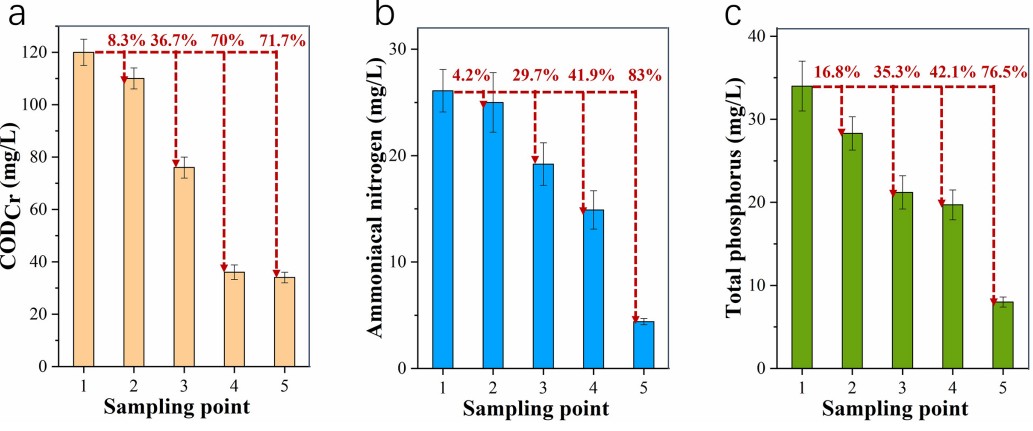

**Figure 5** **Pollutant level changes in the treatment system.** (A) CODCr; (B) ammoniacal nitrogen; and (C) total phosphorus. Red numbers indicate the removal efficiency at different sampling points.

The efficiency of total phosphorus removal was in the order of wetlands >> biochemical tank > flotation tank.

The findings indicate that the biochemical tank plays a pivotal role in the tailwater treatment process. It was the main contributor to $COD_{Cr}$ reduction, achieving a 67.3% removal rate, and significantly aided in the removal of ammoniacal nitrogen (40.4%) and total phosphorus (30.4%). During this phase, organic materials were broken down and oxidized by microorganisms within the activated sludge. Nitrifying bacteria transformed ammoniacal nitrogen into nitrate nitrogen in the presence of oxygen, while phosphorus was absorbed by phosphate-accumulating bacteria and settled into the sludge under aerobic conditions. According to *Liu et al. (2021)* in a multi-stage system involving three ponds and two dams, the aeration pond was the largest contributor to $COD_{Mn}$ (18.7%) and $NH_4^+$-N (28.7%) removal, and it was the second most important for total nitrogen reduction (67.3%) but had a limited effect on phosphorus removal (3.7%). Aeration boosts microbial activity, enhancing the breakdown of organic pollutants, and studies on constructed wetlands treating mixed wastewater have shown strong links between aeration levels and organic matter reduction (*Jianmin et al., 2011*). However, while sufficient aeration ensures the complete oxidation of organic compounds by microorganisms (*Sirianuntapiboon & Jitvimolnimit, 2007*), excessive aeration can destabilize or deactivate the microbes, reducing their efficiency in breaking down pollutants (*Li et al., 2011*).

In the tailwater treatment system developed in this study, the wetland unit played a crucial role, alongside the biochemical tank, in effectively removing ammoniacal nitrogen (70.5% reduction) and total phosphorus (59.4% reduction). These results align with *Liu et al. (2021)*, who found that ecological pools contributed significantly to the removal of total nitrogen and phosphorus. Nutrient removal using aquatic vascular plants was first explored by *Boyd (1970)*. *Ansari et al. (2017)* focused on treating aquaculture wastewater using microalgae. *Enduta et al. (2011)* demonstrated that water spinach achieved 79–87%,

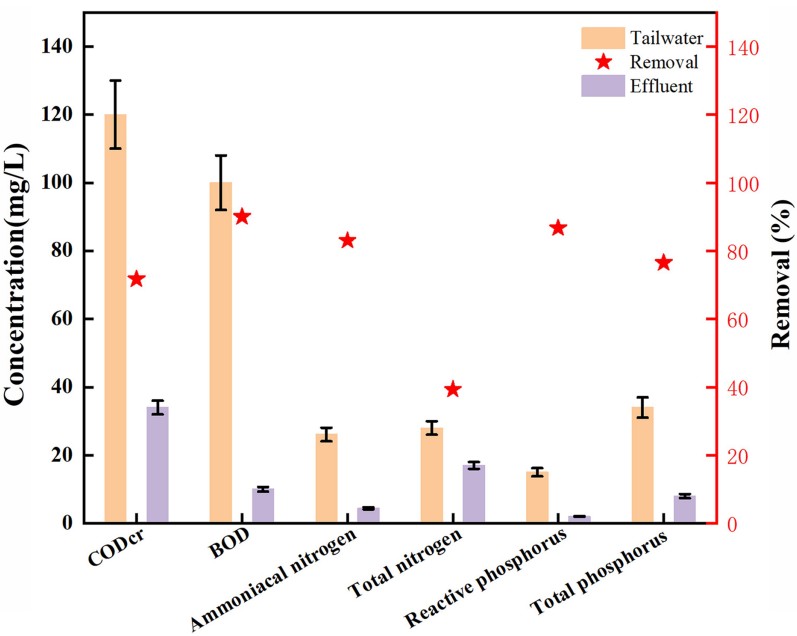

**Figure 6  Tailwater treatment results.**     

75–85%, and 78–85% reductions in nitrate, phosphorus, and total ammonia nitrogen, respectively, in catfish farming wastewater. *Prabhath et al. (2022)* found that Spirulina (*Arthrospira*) platensis removed up to 97% $NH_4^+$-N, 72% $NO_3^-$-N, 96% $NO_2^-$-N, and 93% $PO_4^{3-}$-P from aquaculture wastewater. The microalga *Haematococcus pluvialis* was also shown to significantly decrease $NH_4^+$-N, $NO_2^-$-N, and $NO_3^-$-N levels in a whiteleg shrimp (*Litopenaeus vannamei*) culture system (*Wang et al., 2022*). In our study, watermilfoil, water spinach, and alligator weed, used in the constructed wetlands (Fig. 3), exhibited strong growth and high efficacy in removing ammoniacal nitrogen and total phosphorus. After a month, watermilfoil grew from 5 cm to 40–50 cm in height, with a density increase from 30 to 1,000 plants per square meter. Both alligator weed and water spinach were harvested multiple times, demonstrating their suitability for wetland environments.

## Performance evaluation of tailwater treatment system

The values of all parameters for the tailwater and treatment system effluent are detailed in Fig. 6 and Table 3. Figure 6 highlights the changes in key water quality indicators, including $COD_{Cr}$, biochemical oxygen demand ($BOD_5$), total nitrogen, total phosphorus, ammoniacal nitrogen, and reactive phosphorus. After one month of operation, the treatment system significantly improved water quality, achieving removal efficiencies of 71.7% for $COD_{Cr}$ (reducing from 120 to 34 mg·L$^{-1}$), 90% for $BOD_5$ (from 100 to 10.0 mg·L$^{-1}$), 39.3% for total nitrogen (TN) (from 28 to 17 mg·L$^{-1}$), 76.5% for total phosphorus (TP) (from 34 to 8.0 mg·L$^{-1}$), 83% for ammoniacal nitrogen (from 26.1 to 4.4 mg·L$^{-1}$), and 86.7% for reactive phosphorus (from 15.0 to 2.0 mg·L$^{-1}$). Table 3 presents additional improvements in water quality parameters. Suspended solids were reduced

**Table 3 Tailwater treatment results.**

| Parameter | Tailwater | Effluent | Removal (%) | Grade 1 wastewater discharge standard[*] |
|---|---|---|---|---|
| pH | 6.92 ± 0.12 | 7.21 ± 0.14 | – | 6–9 |
| Suspended solids (mg·L$^{-1}$) | 1,420 ± 20 | 70 ± 2 | 95.1 | 70 |
| Dissolved oxygen (mg·L$^{-1}$) | 1.2 ± 0.1 | 3.5 ± 0.2 | – | – |
| COD$_{Cr}$ (mg·L$^{-1}$) | 120 ± 10 | 34 ± 2 | 71.7 | 60 |
| BOD (mg·L$^{-1}$) | 100 ± 8 | 10.0 ± 0.7 | 90 | 30 |
| Ammoniacal nitrogen (mg·L$^{-1}$) | 26.1 ± 2.0 | 4.4 ± 0.3 | 83 | 15 |
| Nitrite nitrogen (mg·L$^{-1}$) | 0.34 ± 0.03 | 0.027 ± 0.008 | 92 | – |
| Total nitrogen (mg·L$^{-1}$) | 28 ± 2 | 17 ± 1 | 39.3 | – |
| Reactive phosphorus (mg·L$^{-1}$) | 15.0 ± 1.2 | 2.0 ± 0.1 | 86.7 | 0.5 |
| Total phosphorus (mg·L$^{-1}$) | 34 ± 3 | 8.0 ± 0.6 | 76.5 | – |
| Color intensity (degree) | 1,000 ± 30 | 2 ± 1 | – | 50 |
| Odor intensity (level) | 4 ± 1 | 1 ± 1 | – | – |

**Note:**

[*]According to the *Integrated Wastewater Discharge Standard* (GB 8978-1996). P. R. China: State Environmental Protection Administration (*State Environmental Protection Administration, 1996*).

from 1,420 to 70 mg·L$^{-1}$, achieving a remarkable 95.1% removal efficiency, while nitrite nitrogen declined from 0.34 to 0.027 mg·L$^{-1}$ (92% removal). Dissolved oxygen levels increased substantially from 1.2 to 3.5 mg·L$^{-1}$, nearly tripling. Furthermore, the color intensity dropped dramatically from 1,000 degrees to 2 degrees, and odor intensity was significantly reduced from level 4 to level 1, reflecting a substantial enhancement in overall effluent quality.

Significant improvements were observed in color, odor, suspended solids, as well as in the levels of organic matter (COD$_{Cr}$ and BOD$_5$), nitrogen, and phosphorus (Table 3). The treated water met the Grade 1 discharge standard for all parameters, with the exception of reactive phosphorus. Specifically, the concentrations of COD$_{Cr}$, BOD$_5$, ammoniacal nitrogen, and color intensity in the effluent were significantly lower than the national limits, fully complying with Grade 1 national wastewater discharge requirements. The level of suspended solids also met the national standard. However, the reactive phosphorus concentration exceeded the permissible limit, failing to meet the discharge standard. It is important to note that the treated water was not discharged but was reintroduced into the bullfrog farming facility for reuse. Despite the phosphorus levels exceeding the discharge standard, multiple cycles of water reuse showed no adverse effects on the bullfrog farming operations. This indicates that while the phosphorus concentration did not meet the required limit, the system successfully mitigated its environmental impact through the reuse process.

In the treated effluent, despite achieving removal efficiencies of 83% for ammoniacal nitrogen and 92% for nitrite nitrogen, the overall total nitrogen removal was relatively low at 39.3% (Table 3). Studies have indicated that approximately 36% of feed in aquaculture is expelled as organic waste, while roughly 75% of the nitrogen and phosphorus from the feed remain unused and contribute to waste accumulation in the water (*Burford et al., 2003*; *Gutierrez-Wing & Malone, 2006*; *Piedrahita, 2003*). Nitrogen compounds such as NH$_3$,
$NH_4^+$-N, $NO_3^-$-N, $NO_2^-$-N, and organic nitrogen primarily originate from the breakdown of uneaten feed and the waste products of aquatic organisms. In aquatic environments, ammonia ($NH_3$) and ammonium ($NH_4^+$) together are referred to as ammoniacal nitrogen. The removal of ammoniacal nitrogen follows the sequence $NH_4^+$-N→$NO_2^-$-N→$NO_3^-$-N→$N_2$ (*Crab et al., 2007*). Nitrification is a process in which $NH_3 + NH_4^+$ is oxidized and converted to $NO_2^-$-N and $NO_3^-$-N under the action of nitrifying bacteria. Denitrification refers to the process in which $NO_2^-$-N and $NO_3^-$-N are converted to $N_2$ under the action of denitrifying bacteria. Nitrification occurs exclusively in the presence of oxygen, while denitrification requires anoxic conditions and an organic carbon source to act as an electron donor (*Lin et al., 2009*). Nitrifying bacteria are sensitive to several environmental factors, such as elevated ammonia and nitrous acid levels, as well as low dissolved oxygen concentrations. Additionally, they are particularly vulnerable to trace amounts of sulfides found in sediments and sludges in aquaculture systems (*Prosser, 2007*). Furthermore, nitrification is most effective at a low carbon-to-nitrogen (C/N) ratio, as higher ratios favor the growth of heterotrophic bacteria, which compete with nitrifiers for oxygen and space. The lower total nitrogen removal compared to ammoniacal nitrogen and nitrite nitrogen is likely due to unfavorable conditions for denitrification, such as insufficient anaerobic conditions in the biochemical tanks and wetlands, as well as a low carbon-to-nitrogen (C/N) ratio, which hindered the final conversion of $NO_3^-$-N to $N_2$. Future optimizations could involve enhancing denitrification by adjusting hydraulic retention time or introducing additional carbon sources to facilitate microbial activity.

In this study, reactive phosphate and total phosphorus removal rates reached 86.7% and 76.5%, respectively. Despite this, the concentration of reactive phosphate remained slightly above the Grade 1 national discharge standard (as shown in Table 3). Reactive phosphate refers to all soluble forms of phosphate ($PO_4^{3-}$-P). Under anaerobic conditions, phosphate-accumulating bacteria in activated sludge release polyphosphates from their cells (anaerobic phosphorus release), whereas in aerobic conditions, these bacteria absorb phosphorus from the water (aerobic phosphorus uptake) and store it as polyphosphate, leading to the formation of phosphorus-rich biological sludge (*Yao et al., 2024*). This process creates a dynamic balance between phosphorus release and absorption in activated sludge. Phosphorus removal is achieved by discharging phosphorus-rich sludge through sedimentation. In this study, sludge removal was not performed during the operation, and the treated water collected in the purified water tank was reused to replenish the breeding pond. This may be one of the factors contributing to the failure of reactive phosphorus to meet the discharge standard. Additionally, the limited removal efficiency of reactive phosphorus could also be influenced by the specific plant species used in the constructed wetland. According to *Liang et al. (2016)*, different plant species in constructed wetlands vary significantly in their ability to remove nitrogen and phosphorus. *Gichana et al. (2019)* observed that in a compact recirculating aquaponic setup, the pumpkin (*Cucurbita pepo*) demonstrated a much higher phosphorus removal efficiency (75.5%) compared to sweet wormwood (*Artemisia annua*) at 47.36% and amaranth (*Amaranthus dubius*) at 40.72%. Meanwhile, *Paolacci, Stejskal & Jansen (2021)* noted that *Lemna minor* exhibited the
highest nitrogen and phosphorus removal rates at the lowest plant density, with the uptake rates calculated based on the water surface area covered by the Lemna fronds. There is potential for further development of integrated multi-trophic wetlands that include a combination of phytoplankton, algae, hydrophytes, and bioflocs (*Paolacci, Stejskal & Jansen, 2021*). Although reactive phosphorus levels in this study exceeded the permissible limit, the treated water was not released into the environment but rather reused within the system, ensuring no ecological impact. Moreover, the slightly elevated phosphorus levels had no adverse effects on bullfrog growth, reinforcing the system's practicality and ecological safety.

These findings demonstrate that the proposed treatment system substantially improved the quality of tailwater from intensive bullfrog farming, meeting the Grade 1 discharge standard for most parameters. The system's ability to treat the water and reuse it within the farming facility highlights its sustainability and potential for resource-efficient aquaculture practices, despite the minor issue with reactive phosphorus. Further optimization of phosphorus removal could enhance overall performance, enabling full compliance with discharge standards while preserving the system's sustainability and resource efficiency. Potential improvements include extending the biochemical reaction time to enhance biological phosphorus uptake, increasing the length of the plant treatment channels to promote phosphorus assimilation by aquatic vegetation, and incorporating additional physical or chemical phosphorus removal processes at the final stage to further reduce phosphorus concentrations. Additionally, we have considered the treatment and reuse of phosphorus-rich sludge. Phosphorus-rich sludge can be repurposed as an agricultural nutrient through composting or anaerobic digestion, promoting circular resource use. Advanced recovery methods like struvite precipitation offer controlled extraction for fertilizer production. Additionally, proper dewatering and stabilization reduce environmental risks, enhancing resource efficiency and minimizing ecological impact.

## Construction of an integrated recycling system for bullfrog farms

We propose incorporating ecological principles related to food chain dynamics into the design of wastewater treatment systems for intensive aquaculture. Instead of focusing solely on contaminant removal—which might inadvertently lead to secondary pollution—the design should prioritize the "utilization" of chemical substances within the system. For instance, the nutrient-rich active sludge from sedimentation tanks can be periodically extracted and repurposed as organic fertilizer. Similarly, ammoniacal nitrogen and soluble phosphates can be effectively captured and utilized within constructed wetlands (see Fig. 1). In these wetlands, organisms that thrive on ammonia or phosphate can absorb these substances, offering economic advantages and potentially being used as carbon sources or organic matter for fertilizer production. Anionic residues, such as nitrate and phosphate, along with other inorganic anions, can be captured and precipitated using cationic bioflocs, which can then be converted into organic fertilizers. By establishing an integrated recycling system for bullfrog farms, sustainability is enhanced through effective waste management and reuse. This approach minimizes environmental impact, reduces operational costs, and embodies circular economy principles by transforming waste into
valuable resources. This not only improves resource efficiency but also contributes to the overall ecological balance within aquaculture operations.

## Scalability and economic feasibility of the treatment system

The primary costs associated with the system include the construction and installation of flotation and biochemical tanks, plant cultivation in the wetland units, and routine maintenance. The total equipment investment is approximately 4 million CNY, with a treatment capacity of 10,000 tons of water per day. The electricity cost for treating one ton of water is around 0.2 CNY, while additional operational costs, including labor, replacement of consumable parts, microbial inoculation, and alkalinity adjustments, amount to approximately 0.5 CNY per ton. The system's modular design enables scalability, allowing for adaptation to different farm sizes, making it a flexible solution for aquaculture wastewater treatment.

Compared to conventional biological treatment methods, which often require energy-intensive aeration, chemical additives, and sludge disposal, this system has lower operational costs due to its reliance on natural biological processes. Traditional activated sludge-based treatment methods typically incur higher electricity costs, often exceeding 1 CNY per ton of treated water, due to continuous aeration and sludge management. In contrast, the integrated flotation-biochemical-wetland system significantly reduces energy consumption by utilizing microbial degradation and plant uptake for nutrient removal.

Furthermore, the reuse of treated water within the bullfrog farming facility not only minimizes freshwater consumption but also reduces overall operational costs. Additionally, the selected aquatic plants, such as water spinach, have commercial value and can be harvested and sold, providing an additional revenue stream. These factors collectively enhance the economic feasibility of the system, making it a cost-effective and sustainable alternative to conventional wastewater treatment approaches in aquaculture.

## CONCLUSIONS

In this study, a comprehensive treatment system was developed, incorporating a flotation tank, a biochemical tank, and constructed wetlands as key components to manage the tailwater generated from intensive bullfrog farming. The system proved effective in substantially enhancing the quality of the effluent, with most water quality parameters meeting the Grade 1 national wastewater discharge standard. The purified water was suitable for recycling and reuse within the bullfrog farming process, promoting the conservation of water resources and reducing environmental impact. This approach not only optimizes resource use but also offers a scalable model that could be adapted for tailwater treatment in other forms of intensive aquaculture, providing a sustainable solution for water management in the aquaculture industry. To enhance the system's performance, future research should explore alternative phosphorus removal methods, such as advanced chemical precipitation, bio-based adsorption, or integrating membrane filtration technologies, to achieve more efficient and consistent removal. These refinements would further enhance the treatment efficiency and broaden the system's applicability for sustainable aquaculture water management.

### Funding

This study was funded by the Research Project of Fujian Provincial Collaborative Innovation Center for Intelligent Fishery of Higher Vocational College (XTZX-ZHYY-1910), and the Natural Scientific Research Project of Xiamen Ocean Vocational College (KYZ202208). The funders had no role in study design, data collection and analysis, decision to publish, or preparation of the manuscript.

### Grant Disclosures

The following grant information was disclosed by the authors:
Intelligent Fishery of Higher Vocational College: XTZX-ZHYY-1910.
Xiamen Ocean Vocational College: KYZ202208.

### Competing Interests

Chang Sheng Xie is employed by Zhangzhou Haizhiwei Biotechnology Co., Ltd. Dandan Xie collaborates on a project with Zhangzhou Haizhiwei Biotechnology Co., Ltd.

### Author Contributions

- Dandan Xie conceived and designed the experiments, performed the experiments, analyzed the data, prepared figures and/or tables, and approved the final draft.
- Jiehua Hu performed the experiments, prepared figures and/or tables, and approved the final draft.
- Liru Lin performed the experiments, authored or reviewed drafts of the article, and approved the final draft.
- Xiaomei Huang analyzed the data, prepared figures and/or tables, and approved the final draft.
- Changsheng Xie performed the experiments, authored or reviewed drafts of the article, and approved the final draft.
- Haibin He analyzed the data, authored or reviewed drafts of the article, and approved the final draft.

### Data Availability

    The raw measurements are available in the Supplemental File.

### Supplemental Information

Supplemental information for this article can be found online at http://dx.doi.org/10.7717/peerj.19390#supplemental-information.

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
