# Peer review of "Enhancing bullfrog farming sustainability: circular water management through effluent treatment"

_PeerJ, doi:10.7717/peerj.19390_

## Round 0.1 · original submission · Minor Revisions

Notwithstanding the fact that the referees' requests for revisions to the article are of a relatively minor nature, they are, nevertheless, of critical importance. It is, therefore, essential that the suggestions are given careful consideration.

Reviewer 1 ·

Basic reporting

Clarity of Language:

The manuscript is generally well-written and uses professional English. However, certain sections, particularly in the Introduction and Results, could benefit from grammatical corrections and improved sentence flow.
Some examples where language refinement is needed include:
Abstract, lines 20–21: Replace “when operated under optimal conditions” with “operated under optimal conditions.”
Introduction, lines 45–47: The sentence structure is slightly convoluted and could be streamlined for clarity.
Context and Literature Review:

The introduction provides a good overview of the challenges in aquaculture wastewater management. However, a more critical discussion of related studies and how this research advances the field would strengthen the manuscript.
For example, the study briefly mentions constructing wetlands and biochemical filtration in aquaculture but lacks a comparative discussion with prior studies (e.g., Mello et al., 2016, and Gichana et al., 2019).
Figures and Tables:

Figures 1–6 are relevant, well-labeled, and of high quality. However, Table 3, which summarizes treatment results, lacks contextual explanation. The authors should explicitly state whether all parameters meet national discharge standards and discuss the implications of any shortcomings.
Data Availability:

The raw data is sufficiently described, and analytical methods are adequately referenced.

Experimental design

Originality and Scope:

The research question is well-defined and addresses a significant gap in the treatment of bullfrog aquaculture wastewater, an underexplored area compared to fish and shrimp farming.
Methodological Rigor:

The experimental design is robust and replicable. Detailed descriptions of the tailwater treatment system, sampling protocols, and optimization of operating conditions (e.g., hydraulic retention time and aeration intensity) are commendable.
One limitation is the lack of justification for selecting specific plant species (e.g., water spinach and alligator weed) in the constructed wetland. The authors should briefly explain why these species were chosen over others.
Ethical Standards:

Ethical considerations are adequately addressed, and there is no indication of ethical lapses.
Areas for Improvement:

While the experimental design is sound, the manuscript lacks an economic analysis. The feasibility of implementing this treatment system on a large scale would benefit from a discussion on costs and operational considerations.

Validity of the findings

The findings are significant and supported by the data. The removal efficiencies for CODCr (71.7%), ammoniacal nitrogen (83%), and reactive phosphorus (86.7%) highlight the system's effectiveness.
However, the low removal efficiency of total nitrogen (39.3%) and the failure to meet discharge standards for reactive phosphorus require further discussion. The manuscript should explain whether these limitations affect the system's overall viability.
Conclusions:

The conclusions are well-supported by the results and align with the study's objectives. However, they would benefit from a brief acknowledgment of the system's limitations and recommendations for future improvements.
Novelty:

The study introduces a novel combination of treatment units for circular water management in aquaculture. Its emphasis on sustainability and water reuse adds value to the field.

Additional comments

Abstract:

The abstract succinctly summarizes the study but could be improved by briefly mentioning the system's limitations (e.g., phosphorus levels exceeding discharge standards).
Introduction:

Provide a more detailed comparison with existing wastewater treatment methods, highlighting the unique contributions of this study.
Results:

Include a discussion on the practical implications of phosphorus-rich sludge and how it could be managed or repurposed to avoid environmental harm.
Add more context to Table 3 by explicitly stating which parameters meet or exceed discharge standards.
Discussion:

The Discussion would benefit from a dedicated section on the scalability and economic feasibility of the proposed treatment system. What are the anticipated costs of implementation, and how do these compare to conventional methods?
Figures and Tables:

Consider adding annotations to Figures 4 and 5 to make the trends more intuitive for readers.
In Table 3, clarify why certain parameters (e.g., reactive phosphorus) failed to meet the Grade 1 standard and propose potential solutions.
Future Research Directions:

Suggest future research to address the limitations of the system, such as exploring alternative methods for phosphorus removal or integrating advanced technologies like membrane filtration.

Reviewer 2 ·

Basic reporting

In this research, the authors presented the design and implementation of an integrated treatment system, composed of a flotation tank, biochemical tanks, and constructed wetlands, aimed at facilitating the reuse of treated effluent. The figures, tables, and results presented by the authors addressed the purpose of this study and provided valuable information on how to improve bullfrog farming wastewater and enhance its reuse. Overall, the manuscript is well-written but missing references for some important statements, especially in the introduction and discussion sections. Other suggestions to improve this manuscript are listed in additional comments.

Experimental design

No comment

Validity of the findings

An important result of this research is presented in Figure 6. The authors should elaborate on the discussion of this result to improve the reader's understanding. All other results are well presented and explained.

Additional comments

References are missing for some important statements made in the introduction
Other than the general pollution issues that are common to most aquaculture, the authors should highlight some specific reasons the government is prohibiting bullfrog farming in some regions.
Materials and methods well written
Line 76: incomplete reference Sipa˙ba-Tavares et al.
Line 102: Removal = (Ci2Co)/Ci×100%. What kind of removal? Removal efficiency?
Line 126-127: Kindly state your reasons for adjusting the hydraulic retention time
Lines 152-158: Please state the software and version used for the statistical analysis.
Line 178: I think this should be Figure 5(a)??????
Line 209 and 221: "According to Liu et al.,” year is missing in reference
Lines 268-277: reference(s) missing
Line 279: What are some of the unfavourable conditions for denitrification that you observed during your study?
Lines 283-289: reference(s) missing
Line 292: Write reference “Liang et al.,” in full. It appears that the provided reference doesn’t match the one in the sentence.

Some references were cited in the text but not listed, e.g.
Liang et al
Liu et al
SEPA 1996
SEPA 2002

Figure 1: Check for spelling “Damage frog collectionc”
Figure 2: figures should be named 2a, 2b and 2c respectively and corrected in the manuscript where necessary
Figure 3: figures should be named 3a, 3b and 3c respectively and corrected in the manuscript where necessary
Figure 6: authors should kindly discuss separately to improve the reader's understanding of the figure
Table 1: include a superscript to indicate what the numbers in bold italics illustrate

---

## Round 0.2 · accepted · Accept

I have reviewed the revised manuscript and your responses to the reviewers’ comments. It is pleasing to see that the revisions have significantly improved the clarity and scientific contribution of the study. The expanded explanations in the methodology and discussion sections, in particular, have enhanced the reliability of the research and its relevance to the existing literature.

Your thorough approach in addressing the reviewers’ suggestions and making the necessary revisions is a positive step toward the publishability of your work. I believe your study makes a valuable contribution to the scientific understanding of circular water management in bullfrog farming.

I will proceed with the editorial evaluation so that the manuscript can move forward in the review process. Thank you for your contributions.

Reviewer 2 ·

Basic reporting

N/A

Experimental design

N/A

Validity of the findings

N/A

Additional comments

N/A